# Future COVID19 surges prediction based on SARS-CoV-2 mutations surveillance

Fares Z Najar[1], Evan Linde[1], Chelsea L Murphy[1], Veniamin A Borin[1,2], Huan Wang[3], Shozeb Haider[3,4], Pratul K Agarwal[1,2]*

[1]High-Performance Computing Center, Oklahoma State University, Stillwater, United States; [2]Department of Physiological Sciences, Oklahoma State University, Stillwater, United States; [3]University College London School of Pharmacy, Pharmaceutical and Biological Chemistry, London, United Kingdom; [4]University College London Centre for Advanced Research Computing, London, United Kingdom

**Abstract** COVID19 has aptly revealed that airborne viruses such as SARS-CoV-2 with the ability to rapidly mutate combined with high rates of transmission and fatality can cause a deadly world-wide pandemic in a matter of weeks (Plato et al., 2021). Apart from vaccines and post-infection treatment options, strategies for preparedness will be vital in responding to the current and future pandemics. Therefore, there is wide interest in approaches that allow predictions of increase in infections ('surges') before they occur. We describe here real-time genomic surveillance particularly based on mutation analysis, of viral proteins as a methodology for a priori determination of surge in number of infection cases. The full results are available for SARS-CoV-2 at http://pandemics.okstate. edu/covid19/, and are updated daily as new virus sequences become available. This approach is generic and will also be applicable to other pathogens.

## Editor's evaluation

This paper details the creation and data behind the website http://pandemics.okstate.edu/covid19/. The authors explore if there is a cause and effect between the detection of unusually increased mutation activity in the genomic surveillance databases and subsequent near-term surges in SARS-CoV-2 case numbers.

*For correspondence:
pratul.agarwal@okstate.edu

## Introduction

Protein and genome sequence analyses identify molecular level changes that enable viral adaptations for increased spread through the host population. Concrete evidence for a direct relationship between specific mutations and increase in rates of infection (and fatality) requires extensive laboratory studies that need significant time. The availability of unprecedented number of SARS-CoV-2 genome sequences is making possible identification of number and types of mutations, which in turn can provide vital knowledge in real time, crucial for decision making by health professionals for medical interventions. We are investigating several different approaches (synonymous, non-synonymous, and non-synonymous/synonymous ratio for the nucleotide sequences [*Zhang et al., 2006*] and conservative or radical substitutions for the amino acid sequences) for using number and types of mutations as a means to predict surge in infections as well as to monitor the changes in critical viral proteins. Recently, such analysis has been reported for single SARS-CoV-2 proteins (*Kistler et al., 2022*). Our approach, however, is based on the whole viral genome analysis and moreover it is performed continually in real time.

## Materials and methods

The SARS-CoV-2 genomic sequences data and the number of COVID19 (*Platto et al., 2021*) sequences are continually obtained from the sources described below. The genomic sequences are carefully filtered for quality control and used for calculations of non-synonymous ($K_a$) and synonymous ($K_s$) mutation rates for each of the 26 proteins separately.

### Data and data sources

Data for the number of reported COVID19 cases was accessed from Johns Hopkins University's Our World In Data project (https://ourworldindata.org/coronavirus-source-data) (*Dong et al., 2020*).

### Genomic sequence data

An in-house pipeline of scripts (using Linux commands) was designed around the eUtils tools (*Nadkarni and Parikh, 2012*) from NCBI in order to download and process the SARS-CoV-2 records from NCBI's GenBank (https://www.ncbi.nlm.nih.gov/genbank/). Briefly, we used esearch and efetch commands to obtain these GenBank records. Search string 'SARS-CoV-2', refined to 'SARS-CoV-2 [ORGN]', was used to download the identified records in the GenBank text format. After workflow optimization, post May 2022, the search process used NCBI's newer datasets and dataformat command-line tools to identify sequences of interest while continuing to use the efetch tool to download records in the GenBank text format. Collectively as of November 21, 2022, a total of 6,468,196 records were searched and a total of 3,126,129 sequences matching the search criterion and passing the quality control steps were used for the results presented here .

### Quality control

Incomplete and ambiguous SARS-CoV-2 genomic sequences and records containing incomplete collection dates were filtered out using the designed pipeline. For the records passing the quality control steps, the nucleotide sequence for each gene was extracted. A non-redundant version of the extracted nucleotide sequences was derived and translated to the cognate amino acid sequences. In the final phase of the pipeline, the accession numbers for each viral isolate with the nucleotide sequences, the associated protein sequences, the collection dates, and the country of collection were stored in SQLite relational database where they were indexed with unique identifiers to allow the retrieval and analysis of any part of the parsed data.

### Frequency of data updates

As of July 2022, the described sources are monitored daily for updates. New data is continually downloaded and used for analysis.

### Alignments and non-synonymous ($K_a$), synonymous ($K_s$) calculations

The translated proteins and nucleotides sequences were aligned using clustal-omega (*Sievers and Higgins, 2014*) and Pal2Nal (*Suyama et al., 2006*) programs to align the codons with their associated amino acids. The resulting alignments were then processed through the program *kaks_calculator* (*Zhang, 2022*) to calculate non-synonymous ($K_a$) and synonymous ($K_s$) values, and their ratio ($K_a/K_s$) which were used to assess the mutational adaptation for each protein. The parameters required for the *kaks_calculator* were based on the maximum-likelihood method derived from the work of *Goldman and Yang, 1994*. The first reported SARS-CoV-2 genomic sequence ('the Wuhan sequence') (*Wu et al., 2020*) was used as a reference for all the $K_a$, $K_s$, and $K_a/K_s$ calculations. We explored the possibility of using other sequence(s) as references (e.g., the previous day or the previous month), however, due to the increasing number of variations available every day, it is difficult to select a representative sequence on an ongoing basis. It was also found that using the Wuhan sequence as a reference provided the most intuitive and interpretable results.

### List of proteins investigated

The number of unique nucleotide sequences observed till date for each of the 26 proteins/open reading frames (ORFs) are listed in *Table 1*. The full results are available on the project website https://pandemics.okstate.edu/covid19/, which are continually updated.

**Table 1.** Number of unique records for the 26 proteins/open reading frames (ORFs).

Total number of quality-controlled SARS-CoV-2 sequences analyzed: 3,126,129 (as of November 21, 2022). Only three proteins showing the most relevant results and one other protein (marked by *) for comparison are depicted in the figures. These proteins are shown in bold.

| Name | Unique records |
| --- | --- |
| Envelope protein | 1314 |
| Membrane protein | 11,338 |
| Nucleocapsid protein | 70,579 |
| Spike protein | 188,166 |
| Non-structural protein 1 (NSP1), leader protein | 11,656 |
| NSP2 | 67,837 |
| NSP3 | 245,627 |
| NSP4 | 31,257 |
| NSP5, 3C-like proteinase | 11,879 |
| NSP6 | 16,479 |
| NSP7 | 1304 |
| NSP8 | 4490 |
| NSP9 | 2848 |
| NSP10 | 2429 |
| NSP11 | 88 |
| NSP12, RNA-dependent RNA polymerase (RDRP)* | 60,575 |
| NSP13, helicase | 35,421 |
| NSP14, 3'-to-5' exonuclease | 28,501 |
| NSP15, endoRNAse | 12,901 |
| NSP16, 2'-O-ribose methyltransferase | 7636 |
| ORF3a | 41,694 |
| ORF6 | 2117 |
| ORF7a | 9312 |
| ORF7b | 1368 |
| ORF8 | 7036 |
| ORF10 | 710 |

## Results

It was found that collective non-synonymous mutations in key proteins of SARS-CoV-2 showed significant increase 10–14 days before the rapid rise in COVID19 cases, particularly related to the surges that occurred after the emergence of Gamma, Delta, Omicron, and BA.5 variants (*Figure 1* and the related *Figure 1—figure supplement 1* with the unnormalized results). At present, over 6.4 million SARS-CoV-2 genome sequences collected all over the world are available from GenBank (https://www.ncbi.nlm.nih.gov/sars-cov-2/), which were used for analysis of 26 SARS-CoV-2 proteins, including the structural (spike, envelope, membrane, nucleocapsid) proteins, non-structural proteins (NSPs), and ORFs. Note, our analysis was performed with the first reported ('Wuhan') SARS-CoV-2 sequence as a reference (*Wu et al., 2020*). In other words, the computed mutations are calculated in comparison to this reference sequence. The reason for an increase in mutations ahead of a surge is the search for adaptation against the acquired immunity (or gain in function) in either a single protein or a combination of proteins. The case of the Omicron variant indicates the development of the most drastic

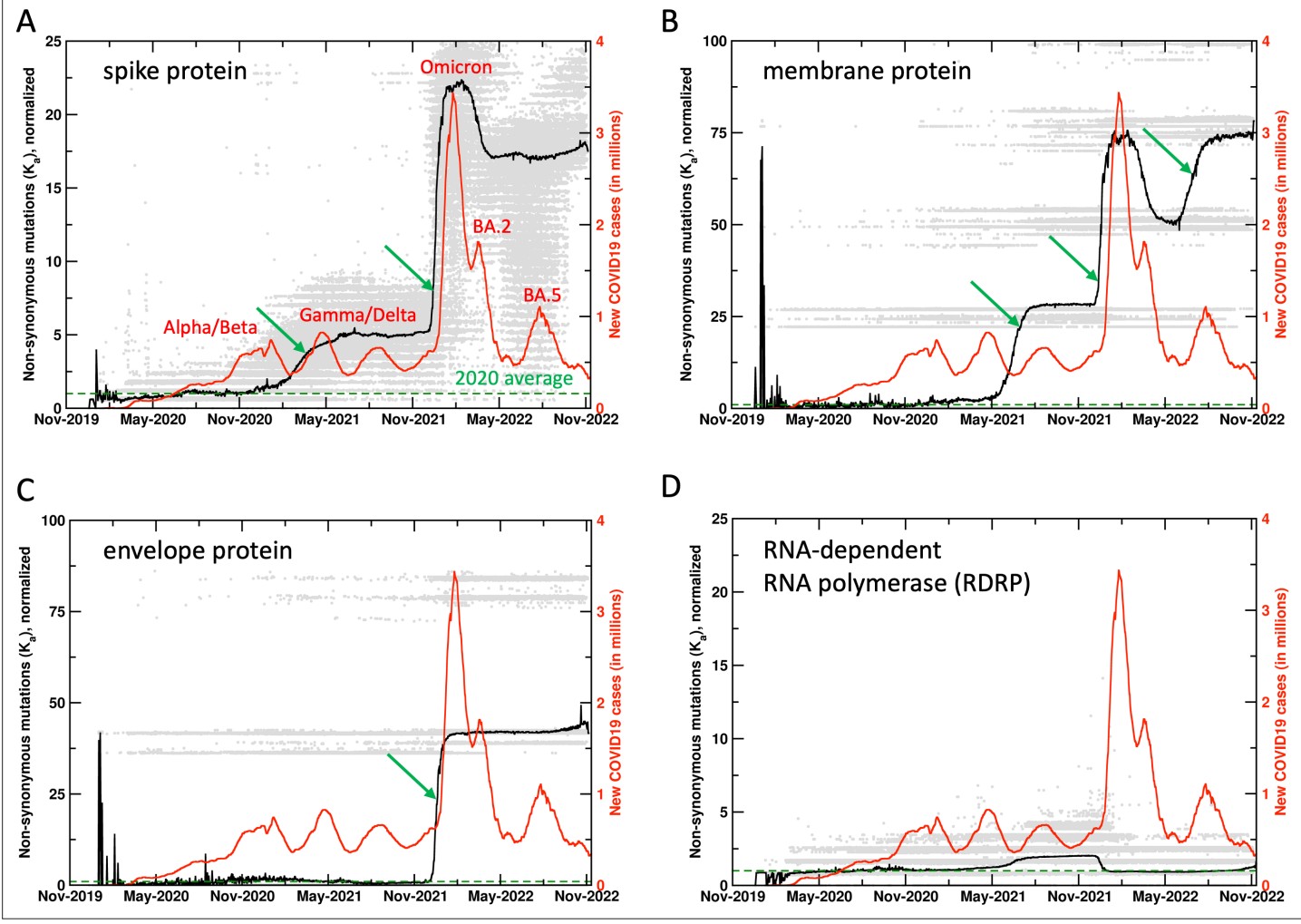

**Figure 1.** Mutations in SARS-CoV-2 proteins increase before COVID19 surges. Non-synonymous mutations over the course of the COVID19 outbreak were identified by analysis of 6.4 million sequences. Gray dots indicate individual mutations, while black lines show weighted means for each day. Red lines show new COVID19 cases (averaged weekly) across the world. The green arrows mark the time when new mutations occurred in significant numbers before the outbreaks, allowing prediction of future outbreaks. The mutation values have been normalized using average of all mutations in the year 2020 (the first full year of the pandemic) as 1 (marked by dashed lines). Raw results are available in *Figure 1—figure supplement 1*. Values of 0 indicate same sequence as the Wuhan sequence, while larger values indicate more mutations. Note that each gray dot corresponds to a unique sequence, and there can be multiple records showing the same mutation. The weighted mean for the day is calculated by using all sequences reported for the day. The peaks for COVID19 cases are labeled with prevalent variants. Alpha/Beta, Omicron, and Omicron BA.2, BA.5 were the prevalent variants at the time of labeled peaks. For the two peaks in 2021 the case was less clear, with Gamma and Delta variants being observed at different times in different parts of the world.

The online version of this article includes the following figure supplement(s) for figure 1:

**Figure supplement 1.** Unnormalized results for the mutations in SARS-CoV-2 proteins.

**Figure supplement 2.** Ratio of non-synonymous mutations/synonymous mutations in SARS-CoV-2 proteins.

**Figure supplement 3.** Daily rate of non-synonymous mutations in SARS-CoV-2 proteins.

**Figure supplement 4.** Side-by-side comparison of various metrics considered in this study.

**Figure supplement 5.** Performance of the surge watch and warning issued on June 29, 2022, and July 14, 2022, respectively.

**Figure supplement 6.** Performance of the surge watch issued on September 7, 2022.

changes in several different proteins, which coincided with the largest increase in rate of infections (*Figure 1*). Non-synonymous mutations ($K_a$) in several proteins show significant increase before the increase in rate of infections (or surges), therefore, allowing a means for surge prediction.

## Use of mutational rates as a surge predictor

In addition to using non-synonymous mutations, a number of other metrics were also investigated for a reliable prediction signal. In particular, the commonly used non-synonymous to synonymous mutations ratio, $K_a/K_s$ (*Figure 1—figure supplement 2*), and the rate of mutations (derivative of observed number of mutations with respect to time) (*Figure 1—figure supplement 3*) were also investigated in detail for suitability as a signal for surge prediction. As seen in *Figure 1—figure supplement 2*, $K_a/K_s$ did not provide a reliable surge prediction signal. *Figure 1—figure supplement 3* shows rate of mutations (calculated as a numerical derivative). For the case of Omicron surge, the proteins did show increased rate of mutations, however, for all other cases a clear signal was absent. Furthermore, the rate of mutations approach presented two additional challenges. First, a number of instances were observed where the rate of mutations increased but did not show increase in reported infections (false positive signal). Second, the nature of incoming genomic data is generally noisy (due to smaller number of samples and weighting of different mutations shows large variations) and changes quickly, therefore, the ongoing most recent rate of mutations is very noisy as well. It was concluded that at this stage, rate (derivative) of mutations is not a reliable signal for surge prediction. In the future, this could be revisited with more stable reporting of genomic sequences with shorter sample collection to sequence publication timeframes. *Figure 1—figure supplement 4* presents side-by-side comparison of the metrics investigated. Overall, it appears that collective non-synonymous mutations ($K_a$) provides the most reliable signal for surge prediction. In the remaining text, we discuss the key results and their importance.

## Spike protein

Spike protein interacts with the angiotensin-converting enzyme 2 receptor and plays a vital role in infecting the human cells (*Xia, 2021*). Spike protein has been the target of mRNA-based vaccines. Viral sequences show significant changes in synonymous and non-synonymous mutations in the spike protein (188,166 unique sequences observed so far), with large increases ahead of the surge in reported human infections, most noticeably with the surges associated with the Gamma/Delta and the Omicron variants (*Figure 1A*). It is important to note that the mutations show increase 10–14 days before the increase in human infections. It is also interesting to note that the synonymous mutations (data available on the website) show decrease post surges. The decrease in mutations prior to the Omicron BA.2 surge corresponds to reversal mutations (returning to reference sequence). However, at present the non-synonymous and synonymous mutations post the Omicron variant remain elevated, more so than any period during the COVID19 outbreak.

## Proteins showing significant mutations

In addition to the spike protein, SARS-CoV-2 membrane (*Lu et al., 2021*; *Figure 1B*, 11,338 unique sequences observed so far) and envelope (*Zheng et al., 2021*; *Figure 1C*, 1314 unique sequences observed so far) proteins have also shown significant mutations, starting just before the Omicron variant (November 2021 onward). For the case of membrane protein, there was a significant increase that started in the Gamma/Delta variants (June 2021 onward) and further increased just before the BA.5 surge. The spike, membrane, and envelope proteins are all located on the surface of SARS-CoV-2 and potentially interact with the components of the immune system. The large increase in mutations in all these external proteins assumes importance in post-vaccination period (discussed further below).

## Other proteins

For comparison, *Figure 1D* shows mutations from RNA-dependent RNA polymerase (RDRP, 60,575 unique sequences observed so far), which has been targeted for development of antiviral drug therapies. Till present, RDRP has shown comparatively lower magnitude of non-synonymous mutations. Note that gray dots are individual mutations, the mean (black line) is weighted by number of sequences for each day by the mutations. Significant increases in mutations are also observed in NSPs 1, 4, 6, 13, 15, ORFs 6, 7a, and 7b (data available on the project website). Overall, this analysis allows us to

monitor ongoing mutations in different proteins; when rapid rise is observed over a short period of time, we issue surveillance watches and warnings (reserved for most extreme cases) for new possible variants with combination of proteins showing new mutations.

## Vaccination and mutational frequencies

Widespread vaccination against SARS-CoV-2 (December 2020 onward) coincides with significant increase in mutation rates of several SARS-CoV-2 proteins. Spike, membrane, and envelope proteins have shown rapid mutations especially in the Omicron variant (gray dots in *Figure 1*). This is possibly due to viral adaptations under the selective pressure exerted by the vaccine, as a significant number of mutations were observed in 2021, especially for the spike protein (gray dots in *Figure 1A* indicate spike protein has mutated much more than any other protein). The long-term effectiveness of mRNA-based SARS-CoV-2 vaccines remains unknown. After the initial regimen of two doses, the administration of additional booster (third and fourth) doses has decreased due to improvement in COVID19 fatality rates as well as political reasons (*Sabahelzain et al., 2021*). This situation raises concerns. Other proteins have shown reversal mutations (higher similarity with the reference sequence) after periods of significant increase in mutations, however, post vaccination the significant mutations observed in the spike, envelope, and membrane protein related to the Omicron variant remain at extremely elevated levels. As Omicron, BA.2, BA.5, and subsequent variants are showing increased rates of transmission, gain or improvement of function in other proteins could lead to emergence of newer variants of concern. Over long term this needs to be addressed by vaccines with longer periods of effectiveness and post-infection treatment options including antiviral drugs.

## Surge prediction

The methodology presented here allows monitoring the potential increase in reported number of human infections. To date, spike protein has shown the most direct correlation in the rate of non-synonymous mutations and the rates of human infection. In particular, in the case of Omicron variant and also the Gamma variant, spike protein showed rapid increase in mutations about 10–14 days ahead of time. Furthermore, membrane protein showed rapid mutations before surge related to BA.5. Therefore, such increase in mutations serves as an indication of upcoming surges. For example, we issued a surge watch on the website on June 29, 2022, which was converted to a warning on July 14. This was confirmed by increase in infection cases worldwide throughout July (see *Figure 1—figure supplement 5*). Further, we issued an additional warning on September 7, 2022, which was confirmed by surge in several European countries, including France, United Kingdom, Germany, and Italy (see *Figure 1—figure supplement 6*).

The role of different (or dominant) SARS-CoV-2 variants in major surges is unclear at this time and needs further research. Different variants have been prevalent in different geographic regions at different times over the course of COVID19 outbreak, therefore, it is difficult to assign the surges to individual variants. In particular, Gamma and Delta variants were both prevalent in different countries in 2021. We are working on enabling this analysis by geographic locations and the results will be available through the website. However, at present our analysis is able to make predictions about collective surges before they occur, as illustrated by the case of BA.5.

In the future, a number of factors could affect the performance of the presented approach. In particular, as the pandemic situation has improved in the second half of 2022, the number of tests being performed and the sequences being deposited into public repositories have decreased. Furthermore, it is widely being discussed that the population is showing increased immunity against the virus due to vaccination and naturally acquired immunity. The presented approach is dependent on availability of sequences, therefore, we hope that scientific community will continue to urge the medical community and public health agencies to commit resources to sequencing the samples from COVID19 positive patients on a regular basis. Nonetheless, even with availability of smaller number of sequences, our approach is weighted by mutations and percentage of sequences showing non-synonymous mutations. Therefore, whenever new mutations show up in large percentages, our approach will still be able to work. On the other hand, viruses continue to evolve and if the population acquires large-scale immunity leading to drastic reduction in number of infections, our surveillance approach would still allow preparation in cases of significant viral genome changes (such as going from SARS-CoV to SARS-CoV-2) whenever they occur and lead to the possibility of another major outbreak.

## Discussion

The methodology and the website described here provide real-time mutational changes of 26 SARS-CoV-2 proteins and ORFs. The changes in non-synonymous mutations correlate with the increase in reported cases of infections. Apart from identifying mutations of concern for in-depth scientific studies, the website is intended to keep the medical community informed about potential upcoming surges. Warnings of increase in mutations and expected surges are displayed on the website (and also available through email alerts). It should be noted that this real-time analysis is dependent on the various health labs and medical facilities for swiftly depositing the viral genome sequences into the public databases such as the GenBank. The shorter the lag time in depositing the sequences by the wider community, more accurate and effective the prediction capabilities of our approach and the website will be.

## Additional information

### Competing interests
Shozeb Haider: Reviewing editor, eLife. Pratul K Agarwal: Founder and owner of Arium BioLabs LLC. The other authors declare that no competing interests exist.

### Funding
No external funding was received for this work.

### Author contributions
Fares Z Najar, Conceptualization, Resources, Data curation, Software, Formal analysis, Supervision, Validation, Investigation, Visualization, Methodology, Writing – original draft, Project administration, Writing – review and editing; Evan Linde, Conceptualization, Resources, Data curation, Software, Formal analysis, Supervision, Investigation, Methodology, Writing – review and editing; Chelsea L Murphy, Conceptualization, Resources, Data curation, Software, Formal analysis, Validation, Investigation, Visualization, Methodology, Writing – original draft, Writing – review and editing; Veniamin A Borin, Resources, Data curation, Software, Formal analysis, Investigation, Visualization, Writing – review and editing; Huan Wang, Formal analysis, Investigation, Visualization, Writing – review and editing; Shozeb Haider, Conceptualization, Supervision, Investigation, Visualization, Methodology, Writing – review and editing; Pratul K Agarwal, Conceptualization, Data curation, Formal analysis, Supervision, Validation, Investigation, Methodology, Writing – original draft, Project administration, Writing – review and editing

### Author ORCIDs
Evan Linde  http://orcid.org/0000-0002-2053-6721
Chelsea L Murphy  http://orcid.org/0000-0001-5367-8593
Shozeb Haider  http://orcid.org/0000-0003-2650-2925
Pratul K Agarwal  http://orcid.org/0000-0002-3848-9492

### Decision letter and Author response
Decision letter https://doi.org/10.7554/eLife.82980.sa1
Author response https://doi.org/10.7554/eLife.82980.sa2

## Additional files

### Supplementary files
• MDAR checklist

### Data availability
All sequences used in this work are available from GenBank. The protocol used for analysis are described in the Materials and methods section.

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
