## [Editor Report]

This paper details the creation and data behind the website http://pandemics.okstate.edu/covid19/. The authors explore if there is a cause and effect between the detection of unusually increased mutation activity in the genomic surveillance databases and subsequent near-term surges in SARS-CoV-2 case numbers.

---

## [Decision Letter]

**Decision letter after peer review:**

Thank you for submitting your article "Future COVID19 surges prediction based on SARS-CoV-2 mutations surveillance" for consideration by *eLife*. Your article has been reviewed by 2 peer reviewers, one of whom is a member of our Board of Reviewing Editors, and the evaluation has been overseen by Mone Zaidi as the Senior Editor. The reviewers have opted to remain anonymous.

Essential revisions:

1) Update the manuscript to include recent prediction/performance data on the Omicron variants.

2) Please address in detail all the recommendations made by the two reviewers.

*Reviewer #1 (Recommendations for the authors):*

The following should be addressed:

The paper notes that the ratio of non-synonymous to synonymous mutations (ka/ks) which is typically used, did not provide clear trends.

Ideally, these would be plotted against the best model that the authors are proposing. Separate graphs fail to demonstrate where one model fails and another succeeds.

The paper has numerous typos and careful proofreading is warranted. For example, line 170 "Therefore, such increase is mutations serve" should read "in"

Line 182 "reported cases of infections. Apart from identifying mutations of concerns for in-depth scientific studies" should read "concern"

Line 183 "the website is intended to keep the medical community informed about the potential surges." should read "about potential" without "the".

Additionally, potentially off-putting text should be rephrased. For example, line 157 "improvement in COVID-19 fatality rates as well as political reasons. This situation raises concerns." should be re-phrased to eliminate politics as a conjecture unless there is a documented reference to such.

*Reviewer #2 (Recommendations for the authors):*

This work suggests an interesting approach to stay ahead of the virological case curve but is too preliminary. The manuscript itself doesn't make any firm conclusions as to which protein or set of proteins is the most predictive. The claim of 14 days is hard to verify from the data provided in Figure 1 and is likely restricted to the detection of the Omicron variant. Furthermore, the method does not do a good job of predicting the variants evolving from the Omicron variant, suggesting that the baseline comparison needs to be refreshed. Given that the authors' focus is on infectivity, perhaps a more targeted focus at receptor binding domains of the spike protein may provide a more robust method, similarly providing comparisons based on major locations of the world (e.g. US, Europe, South Africa, India) might provide cleaner data as it could detect the local emergence of strains. The authors should also discuss the usefulness of such an approach in light of increased population immunity and decline in testing, which will negate the advantage of sequencing over case detection.

---

## [Author Response]

Essential revisions:1) Update the manuscript to include recent prediction/performance data on the Omicron variants.

We have updated the manuscript with the most up to date data available. The data is current up to 21^st^ November 2022, including the Omicron variants. Further, we have included the predictions we made in June and September and their validation based on the increase in cases of infection.

2) Please address in detail all the recommendations made by the two reviewers.

We have made revisions as described in detail below.

Reviewer #1 (Recommendations for the authors):The following should be addressed:The paper notes that the ratio of non-synonymous to synonymous mutations (ka/ks) which is typically used, did not provide clear trends.Ideally, these would be plotted against the best model that the authors are proposing. Separate graphs fail to demonstrate where one model fails and another succeeds.

Thanks for the suggestion. We have included these plots as Figure 1—figure supplement 4.

The paper has numerous typos and careful proofreading is warranted. For example, line 170 "Therefore, such increase is mutations serve" should read "in".

Done. Thanks for pointing this mistake out. We have carefully proofread the manuscript and corrected such mistakes and other errors.

Line 182 "reported cases of infections. Apart from identifying mutations of concerns for in-depth scientific studies" should read "concern"

Done.

Line 183 "the website is intended to keep the medical community informed about the potential surges." should read "about potential" without "the".

Done. Thank you again.

Additionally, potentially off-putting text should be rephrased. For example, line 157 "improvement in COVID-19 fatality rates as well as political reasons. This situation raises concerns." should be re-phrased to eliminate politics as a conjecture unless there is a documented reference to such.

Done. We have added appropriate reference (Sabahelzain, MM, Hartigan-Go, K, Larson, HJ. The politics of COVID-19 vaccine confidence*. Current Opinion in Immunology*. 2021;71;92-6.). We hope that the reviewer finds our changes acceptable.

Reviewer #2 (Recommendations for the authors):This work suggests an interesting approach to stay ahead of the virological case curve but is too preliminary. The manuscript itself doesn't make any firm conclusions as to which protein or set of proteins is the most predictive. The claim of 14 days is hard to verify from the data provided in Figure 1 and is likely restricted to the detection of the Omicron variant. Furthermore, the method does not do a good job of predicting the variants evolving from the Omicron variant, suggesting that the baseline comparison needs to be refreshed. Given that the authors' focus is on infectivity, perhaps a more targeted focus at receptor binding domains of the spike protein may provide a more robust method, similarly providing comparisons based on major locations of the world (e.g. US, Europe, South Africa, India) might provide cleaner data as it could detect the local emergence of strains.

We thank the reviewer for very insightful feed-back. Like the reviewer we are extremely focused on rigor, validation and reproducibility of our approach. The reviewer mentions a number of interesting points. Our responses (in sequential order) are provided below:

1. Regarding the approach being too preliminary, as mentioned in response to the other reviewer, our approach has been successful as prediction issued in September (while this manuscript was in review) was validated by surge in number of cases in several European countries. We would very much like to immediately validate this approach more thoroughly, however, unlike the lab studies which could be addressed by additional experiments, we are dependent on the incoming data with the evolution of this pandemic. Our warnings from June and September have been validated and we are confident that this approach and website will help the medical and research community. We will continue to refine the approach.

2. Conclusion about which set of proteins: Based on the data so far, we do not believe that a single protein or few proteins will cause new surges (as discussed in the manuscript). Analysis reveals that any protein with new mutations could cause surges. Therefore, our website presents data on all SARS-CoV-2 proteins. We are concerned regarding making conclusions based on a limited set of proteins, as this could be detrimental to the cause and possibly miss the surges driven by other proteins.

3. 14-days: Please see the plots in response to Reviewer 1. We issued a warning on September 7^th^ and the cases started rising in the middle to late September. We believe additional surges would provide us more confidence.

4. Limited to Omicron variant: We believe the reason for this is that a lot more sequences became available. It is not a limitation of the approach but rather a limitation due to the availability of data.

5. Predicting the evolution of variants: This is a very interesting point. If we understood this comment correctly, reviewer is asking if we could predict the emergence of new variants. Even though it is very much possible but prediction of new variants (through mutational adaptations) was not the focus of the current study. But we will keep this suggestion in mind for ongoing studies. However, if the comment was regarding making surge predictions for Omicron sub-variants, new data has now been included in the revised manuscript. We thank the review for this suggestion.

6. Baseline comparison needs to be refreshed: If we understood correctly, the suggestion is to change the reference sequence from the Wuhan sequence to another appropriate sequence. We have been wondering about this ourselves (as noted in the original manuscript). However, the question is which one? The answer is not clear to us immediately as there are multiple variants prevalent across the world. We are keeping an eye on literature and if there is an obvious candidate, we will in future shift to a new reference sequence. In the meantime, we are also exploring the use of different reference sequences for different countries.

7. Receptor binding domains: As mentioned above, the approach is meant to be general and not tied to a particular protein. We are afraid that focusing on a single protein or a domain will miss out on the future breakouts. We continue to list more interesting mutations on all SARS-CoV-2 proteins in our warnings. If data suggests that one single protein or a domain is highly indicative of the surges in the future, we will publish an update.

8. Different locations and countries: This is a great point! As per reviewer’s suggestion we have enabled the infrastructure on our website. We have listed the charts associated with the US on the website (under the section More Charts in the top navigation menu). However, we found that the GenBank does not have sufficient number of sequences from other countries to provide the resolution needed. Therefore, we are working on enabling the real time analysis of the GISAID data. However, we found that the sequences in GISAID vary extremely widely in quality. Since we are aiming for high quality and reproducible studies (as both reviewers also suggest), we do not feel comfortable at this stage in including the results for different countries in the main manuscript. But reviewer’s point is a great one and we have built the infrastructure for geographic break-down analysis and the results will be continually added to the project website once quality control is considered satisfactory.

The authors should also discuss the usefulness of such an approach in light of increased population immunity and decline in testing, which will negate the advantage of sequencing over case detection.

Done. This is another interesting point. We have added a new paragraph in the manuscript at the end of Results section. Briefly, as widely discussed in the community, the immunity (across the population) is being developed. However, new variants keep on emerging. Therefore, it is important to monitor new mutations in all proteins. The decrease in testing is not optimal. We are hopeful, our results will encourage medical community and public policy makers to continue testing. In the meantime, we believe the ratio of sequences showing new mutations compared to no mutations for the same day (or the same week) will continue to be useful, even though the overall sequences being reported may go down.

Once again we thank the reviewer for all the insights and great suggestions.